# Location Privacy Protection Systems in Presence of Service Quality and Energy Constraints

**Mulugeta Kassaw Tefera**  **and Xiaolong Yang \***

School of Computer and Communication Engineering, University of Science and Technology Beijing, Beijing 100083, China; mulugetaksw@gmail.com
**\*** Correspondence: yangxl@ustb.edu.cn

**Abstract:** The wide-ranging application of location-based services (LBSs) through the use of mobile devices and wireless networks has brought about many critical privacy challenges. To preserve the location privacy of users, most existing location privacy-preserving mechanisms (LPPMs) modify their real locations associated with different pseudonyms, which come at a cost either in terms of resource consumption or quality of service, or both. However, we observed that the effect of resource consumption has not been discussed in existing studies. In this paper, we present the user-centric LPPMs against location inference attacks under the consideration of both service quality and energy constraints. Moreover, we modeled the precision-based and dummy-based mechanisms in the context of an existing LPPM framework, and also extended the linear program solutions applicable to them. This study allowed us to specify the LPPMs that decreased the precision of exposed locations or generated dummy locations of the users. Based on this, we evaluated the privacy protection effects of optimal location obfuscation function against an adversary's inference attack function using real mobility datasets. The results indicate that dummy-based mechanisms provide better achievable location privacy under a given combination of service quality and energy constraints, and once a certain level of privacy is reached, both the precision-based and dummy-based mechanisms only perturb the exposed locations. The evaluation results also contribute to a better understanding for the LPPM design strategies and evaluation mechanism as far as the system resource utilization and service quality requirements are concerned.

**Keywords:** location-based services; location privacy; location-privacy preserving mechanism; resource utilization; tracking attacks; game-theoretic model

## 1. Introduction

Recent advances in mobile computing and wireless communication technologies take advantage of mobile applications to provide location-based services (LBSs) to mobile users. The increasing usage of LBSs in location-aware mobile devices has significantly benefited users in various aspects of their day-to-day activities such as GPS navigation, finding nearby points-of-interest services, mobile local search, etc. Many convenient services provided by these LBSs are based on the users' geographical position to obtain their real-time location data and contextual information about their surroundings. While LBSs bring great benefits to their users, it also increases the danger of user privacy breaches due to untrusted third-party adversarial applications. The reason for this is that when users conveniently access various LBSs, they need to report their real-time location information and other related service attributes in the communication network. The context attached to this location information contains not only location privacy, but also other sensitive personal data that the user usually wants to protect such as health status, living habits, home address, and social relations [1,2]. However, due to the existence of untrusted third-party LBS applications, the user private information is bound to cause serious privacy

concerns [3–5]. Therefore, the need to protect a user's LBS privacy is a crucial and urgent task in the LBS system. To address this concern, a number of LPPMs have been discussed in the literature to obfuscate the real locations before being reported to the LBS provider. These protection mechanisms are based on generating dummy queries with fake locations or reducing the amount of exposed location information to the LBS provider [6–8]. Commonly used privacy protection strategies include dummy-based LPPMs and precision-based LPPMs [9]. In the case of dummy-based solutions [10–12], the LPPMs are sensitive to a number of fake pseudo-locations together with the user's actual locations; therefore, protecting the real identities of the users is more significant. In contrast, precision-based LPPMs [13–17] are sensitive to reducing the precision of disclosed locations before being reported to the LBS provider, therefore there is a large geographical region where the users might be located.

Furthermore, [18] proposed a design framework for a LPPM that considered the adversary's knowledge on the user's LBS access pattern and the LPPM obfuscation algorithm. This framework enables an adversary to estimate the privacy protection degree, which can be expressed as a distance function between the estimated location and real location. In addition, the design methodology allows users to set a maximum achievable service quality loss resulting from the use of a perturbation-based LPPM. This study was suitable to model the wide use of LBSs where users expose their location information in a sporadic manner, e.g., applications for nearby points-of-interest search services, mobile local search events, and location check-in services. However, while the study was suitable in solving the optimal protection and attack strategies in a game problem, we observed that the problem statement did not consider the effects of resource utilization on privacy protection, (e.g., energy and bandwidth consumption). These are, however, likely to be the key issues for LBSs due to the high cost of resource consumption and communication capabilities using smart mobile devices.

In this paper, we aimed to study a user-centric LPPM against location inference attacks under service quality and energy constraints as an extension of the framework in Reference [18]. We allowed the mobile users to implement local and user-centric LPPMs to generate their real location into a pseudo-location (obfuscated location) before sending it to the LBS provider. Considering the reality that mobile users may not be sufficiently inspired to transform pseudo-locations due to the high cost of pseudonym generation using resource-constrained mobile devices, we applied a game-theoretic model to analyze the optimization problem between the user and the adversary, and then identified the equilibrium solutions. Based on this, we analyzed the privacy protection effects of optimal location obfuscation and attack strategies in order to achieve the optimization problems under several real mobility datasets. Furthermore, we introduced two other popular solutions such as the dummy- and precision-based LPPMs to achieve the effective protection of a fine-grained comparative evaluation. We formalized the optimization problems based on the existing perturbation-based framework that generated the real location before being revealed to the service provider. As both dummy and precision-based protection mechanisms only perturb the previously exposed locations, each pseudo-location outputted by the LPPMs may or may not include the user's real location, which results in the consumption of extra resources. To model the optimal privacy protection and attack strategies within the LPPMs, we extended the existing perturbation-based optimization problems that are optimal against each other.

Therefore, the main important contributions (advantages) of this paper can be abstracted as follows. First, we extended an existing framework [18] for protecting location privacy of users in the LBS to ensure the satisfaction of the quality of service and energy consumption. Second, we modeled the optimal precision-and dummy-based mechanisms that could be incorporated in the context of the existing framework, thus providing better privacy protection performances. Relying on the existing optimization problems, we formulated the linear program solutions for these popular protection mechanisms under several real mobility datasets. Third, the evaluation methods for different LPPMs considered the successively visited locations to effectively protect the location privacy of users along their access to the LBSs. When the adversary knows the users' choice of LPPMs, then the users radically change the way that they want to be protected when they use the LBS.

The rest of this paper is organized as follows. Section 2 presents the related works and our motivation. Section 3 introduces the formal location-privacy framework and the basic assumptions around the system architecture. Section 4 describes the problem statement and the privacy game formulation to quantify location privacy against inference attacks. The performance evaluation and experimental settings against real location traces are presented in Section 5. Finally, we present our discussion and conclusions in Section 6.

## 2. Related Work

Protecting the user's location privacy is a basic requirement for the successful deployment of LBS applications [19]. The most popular protection mechanism to obtain location privacy is to report a fake-location or obfuscated-version of their location information to the service provider [20,21]. These obfuscation mechanisms act as a noisy information channel between the LBS provider and the user's actual location with the intention of confusing an adversary [22]. According to Reference [23], when users want to protect their location privacy in LBSs, they use an obfuscation mechanism before the information is released to an untrusted observer (e.g., using location hiding and perturbation). Thus, hiding the user's actual location can be done by using misleading information and replacing it with false or pseudo-locations. In a perturbation-based mechanism, a user's reported location can be implemented (i.e., replaced with other fake-locations), for example, by adding a noise channel to the user's real location. The different protection mechanisms can be combined by hiding the user's location using mix-zones, which allow users to hide their private information [24], and further reduce the communication between the adversary and service provider. Other LPPMs allow users to automatically generate multiple fake-locations (pseudo-locations) together with the user's actual whereabouts that are indistinguishable from the users' real locations [10–12]. In this case, the adversary is unable to estimate whether the decoy locations or associated pseudo-locations correspond to the user's exact positions. Finally, precision-based mechanisms [13–17] are sensitive to minimizing the granularity of the location information disclosed to the LBS observer, and therefore it is impossible for the adversary to identify the user's real location within a geographical area of a region.

Recently, significant research works have discussed formalizing the desirable location privacy requirements and finding an appropriate evaluation metric to measure the privacy protection performance. For example, Reference [25] proposed an important method for modeling and evaluating the user's location privacy between the estimated location and actual location using distortion-based metrics. Thereafter, similar, but more general uses of location privacy measurement methods have been further proposed in References [18,26,27] to quantify LBS privacy. These measured the location privacy leakage as the adversarial expected error when estimating the user's actual location by considering the obfuscated location and prior knowledge. This quantitative evaluation method was the basis of their design framework for the LPPM, where they provided a systematic design method for an optimal protection strategy against smart adversaries who could understand the users' LBS access pattern and their choice of protection mechanism at the given time instant. This design framework enables mobile users to achieve the maximum achievable service quality and location privacy that they are always willing to trade-off. To the best of our understanding, this design framework is an effective LPPM that explicitly includes the adversarial model into the design consideration that accounts for common information (knowledge) between the attacker and the privacy protector. We note that our work builds on this analytical framework and extends it to ensure the satisfaction of quality of service and resource limitations on energy consumption. The proposed framework by Reference [18] did consider an adversary that had information on the users' access to the LBS and the LPPM's internal algorithm in order to measure location privacy. Nevertheless, their work did not capture the constraints on the resource consumption (e.g., energy constraint) implemented by some privacy preserving strategies such as decreasing the precision of disclosed locations or generating dummy queries.

In this work, we used a different approach to study the existing user-centric privacy protection strategy to specify the resource limitations. The conflict between user privacy and service quality

with the objectives of the adversary has been discussed in the literature [18]. In the location-privacy scenario, the game theory model is an advanced tool for analyzing the user's privacy protection problems in various LBSs [6,28]. In this paper, we extended the classic formulation of the game theory model to better understand the problem of optimal privacy protection and attack strategies. A similar approach was used by Reference [29] in the extension of the existing framework, taking into account the bandwidth constraints to comparatively evaluate the different LPPMs. Our solution is unlike previous work on protecting the users' privacy while ensuring the satisfaction of the quality of service and energy consumption.

## 3. Privacy-Preserving System Framework

This section first describes a formal block diagram for a privacy protection scenario and then explains the user and adversary considerations, the privacy-preserving mechanisms, and privacy evaluation metrics, respectively. We considered the local and user-centric protection analysis in which the modifications are made individually by the users or on their mobile devices without the need of other users' knowledge in the privacy protection system. Therefore, we limited the implementation and modeling to a single user (i.e., in a user-centric manner) in the rest of our analysis in this work. Figure 1 illustrates the resulting block diagram for the user-attacker scenario and the interactions between different components [23,30].

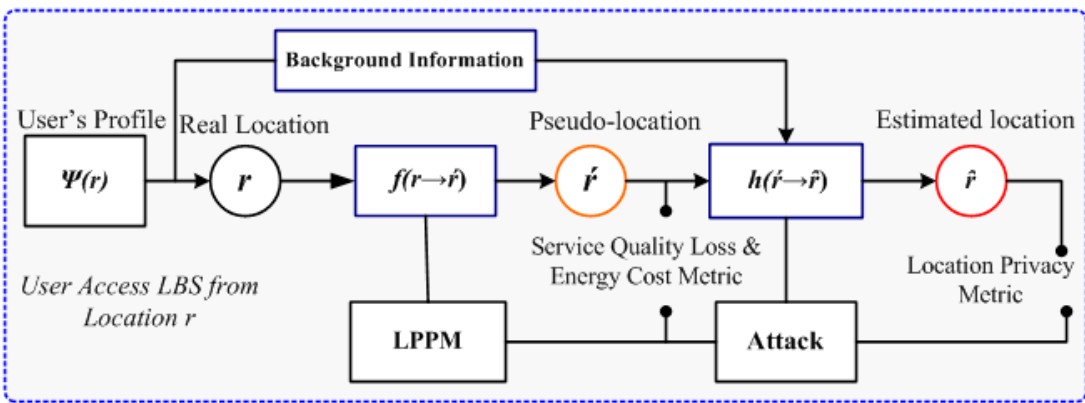

**Figure 1.** A block diagram for the user–attacker privacy protection scenario.

As shown in the figure, the system framework contains three main actors: (i) The LBS users, who are willing to disclose their real location information while becoming more aware of their privacy when accessing LBSs; (ii) the trusted privacy preserving system (i.e., LPPMs), which utilizes privacy enhancement techniques to guarantee the privacy of the LBS users; and (iii) an untrusted service provider, who ignores the interest of the LBS users or are reluctant to disclose their private information. This framework allows us to specify how user mobility is represented in our user-centric analysis and then evaluate the effectiveness of the corresponding LPPMs against location inference attacks. Next, we describe our assumptions regarding the adversary model and privacy protection mechanisms. In Table 1, we define the basic notations introduced in the rest of the section.

**Table 1.** Summary of notations.

| Symbol | Meaning |
|---|---|
| $r, \acute{r}, \hat{r}$ | User's real-location, observed (pseudo-location), and an adversary's estimate location, respectively. |
| $R, \acute{R}$ | Set of real and pseudo-locations, respectively. |
| $\psi(r)$ | Probability of accessing LBS from location $r$, (user's profile). |

**Table 1.** *Cont.*

| Symbol | Meaning |
|---|---|
| $f(\acute{r}\backslash r)$ | The user's location privacy-protection function to transform $r$ as $\acute{r}$ at observation time $t$. |
| $h(\hat{r}\backslash\acute{r})$ | The probability attacks function for adversary to estimate location $\hat{r}$ closing to the real-location of the users under their observed location $\acute{r}$. |
| $d_q(\acute{r}, r), d_E(\acute{r}, r)$ | Distance error functions between the real-location $r$ and pseudo-location $\acute{r}$, which determines the user's LBS loss of service-quality and energy constraints. |
| $d_p(\hat{r}, r)$ | Distance error function between the location $\hat{r}$ estimated by the attacker, and the user's real location $r$, which can measure the location privacy leakage of the user, or equivalently the adversary's expected error. |
| $Q_{loss}(\psi, f, d_q)$ | Function of expected service-quality loss, given the user's mobility profile $\psi(r)$ and an obfuscation function $f(r)$ at time $t$. |
| $E_{cost}(\psi, f, d_E)$ | Function of expected energy cost, given the user's profile $\psi(r)$ and location obfuscation function $f(r)$ at time $t$. |
| $Q_{loss}^{max}$ | Maximum achievable loss quality of service. |
| $E_{cost}^{max}$ | Maximum tolerable energy cost imposed by users. |
| $Privacy(\psi, f, h, d_p)$ | User's expected location privacy with mobility profile $\psi(r)$ using obfuscation function $f(r)$ against inference attack function $h(r)$. |
| $Pr$ | Represents the probability density functions (PDFs), which are sub-indexed by the corresponding random variable in case of uncertainty. For example, $Pr(\acute{r})$ denotes the probability value of function $Pr$ at $\acute{r}$ and $Pr(r\backslash\acute{r})$ denotes posterior probability values for $r$, given a status of nature in $\acute{r}$. |

### 3.1. User and Adversary Consideration

The objective of the adversary is to construct the location access profile of the users and then infer their private information at a given time. Here, the adversary might be a service provider or any entity that can capture private information on the users' LBS access. We considered the situation where $N$-mobile users in the LBS move in a geographical area partitioned into $M$-discrete regions $R = \{r_1, r_2, \ldots, r_M\}$. Each area of region acts as the minimum size of locations when users move from one place of region to another for different purposes (such as for shopping, work, etc.). In this setting, mobile users report their location information and other contextual information to the LBS at each time instant. We assumed that the adversary was knowledgeable to the type and characteristics of the LBS, and could observe all of the distortion pseudo-location outputted by the LPPMs. When the adversary knows the users' LBS profile and the logic of their protection mechanisms, it will use that knowledge to update their attacking strategy and therefore improve the success rate of attack. According to Reference [18], the effectiveness of the adversary's strategy can be measured as the probability attack function $h(\hat{r}\backslash\acute{r}) = Pr(\hat{r}\backslash\acute{r})$, which defines the probability of the attacker's estimated location $\hat{r}$ close to the actual-location of the users under their exposed location $\acute{r}$. In nature, the user's privacy leakage is in inverse proportion to the adversary's expected-error between its reconstructed location $\hat{r}$ and the user's actual-location $r$.

Hence, the adversarial estimation error can be formulated as a function related to $d_p(\hat{r}, r)$, which greatly relies on the user's privacy requirements, and location semantic diversity (user's sensitive information). This is because each user's location data contain their sensitive private information (semantic service-attribute), therefore, the user needs to protect them, e.g., the status of the user's health condition in hospital. In order to access a location-related service, users are required to sporadically share/obtain their exact current position with a LBS provider.

Let $\psi(r)$ be the user's profile, as it considers the various pieces of information regarding the users' access to the LBS or (the probability of accessing the LBS at time $t$ from location $r$). As seen from the attacker's side, $\psi(r)$ is the probability of accessing the LBS from location $r$, $\sum_r \psi(r) = 1$, for any time

instant *t*. This can be computed by the adversary from the successively observed locations of users and from the location access pattern of their LBSs in the given time. Each mobile user potentially uses their location enabled devices to report local search services or other queries at some frequency to the LBS. We assumed that the user accesses their LBS profile, but does not make any particular restrictions on his profile $\psi(r)$. As the users are on their smart phones, their moving location is a time-dependent value (e.g., the user's access patterns may be different in the morning than in the afternoon). Therefore, the LBS profile of the users describes the frequency of their visits to particular locations, and does not contain any assumptions about the transitions between locations. We note that $\psi_t(r)$ is the prior-information for the adversary on the user's LBS access at time *t*, before the adversary observes the user's real location traces.

### 3.2. General Considerations for LPPMs

We considered the mobile users who are willing to disclose their location information while preserving their private information from malicious attackers who can observe the users' locations and background information. The LPPM receives a set of the users' real location *r* (*r* ∈ *R*), and then produces an appropriate pseudo-location *ŕ*, (*ŕ* ∈ *Ŕ*) before sharing it with the LBS provider. We note that the focus is on local and user-centric protection strategies where the transformation decisions are made separately from each other without the need of a third-party LBS provider that represents a location server (anonymizer). Hence, when mobile users conveniently access LBSs, their location and service attributes only reveal the output of the LPPM, rather than sharing their real-location traces, and at the same time, the adversary can reconstruct/infer it by the observed (pseudo-location) *r'*. For the sake of completeness, we set *R'* = *R*, nevertheless, in the most common sense, *R'* is the power set of *R* and has the same attribute, which corresponds to a candidate set of available location-traces (regions) in *R*.

The LPPMs based on dummy and precision mechanisms not only transform decoy-locations, but also obfuscate the users' previously observed pseudo-locations [9]. Each obfuscated-location *ŕ* corresponds to the *R*, (i.e., *ŕ* ∈ *Ŕ*) output by LPPM, which might not include the user's actual location (i.e., $r_k$ ∈ *R*), and might be produced by multiple successive regions from the mobility profile $\psi(r)$. Let us consider Figure 2a as an example to explain the comparison of different protection mechanisms, where the point of interest area *R* is produced by 16 regions. In this setting, the user queries three service requests and accesses the LBS from locations $r_5$, $r_{10}$, $r_{15}$ (i.e., $r_{t,3}$ = {$r_5$, $r_{10}$, $r_{15}$}). In Figure 2b, we can observe that two possible pseudo-location traces $ŕ_{t,3}$, are depicted in the white and gray background.

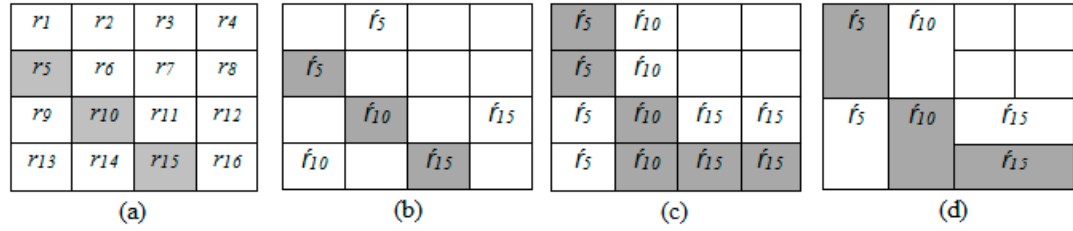

**Figure 2.** Comparison of the differences between various LPPMs: (**a**) Real location trace, (i.e., *R* = {$r_1$, ..., $r_{16}$}), (**b**) perturbation-based mechanism (i.e., *R* = *Ŕ*), (**c**) dummy-based mechanism, and (**d**) precision-based mechanism.

Note that the gray background of each pseudo-location $ŕ_{t,3}$ coincides with the user's real location $r_{t,3}$ = {$r_5$, $r_{10}$, $r_{15}$}, while the white background pseudo-locations $ŕ_{t,3}$ do not. In both dummy and precision mechanisms, the real location traces *R* = {$r_1$, ..., $r_{16}$} of each of the exposed events can be expanded to a certain range. As shown in Figure 2c, each real location can be replaced by two possible pseudo-locations, where one location-dummy is produced by two other non-contiguous locations. In Figure 2d, the expanded interest area of each gray background pseudo-location, i.e., {$ŕ_5$, $ŕ_{10}$, $ŕ_{15}$} is equal to {{$r_1$, $r_5$}, {$r_{10}$, $r_{14}$}, {$r_{15}$, $r_{16}$}} which includes the actual location traces, while that of the white color background pseudo-location does not. However, the transformation decision of each LPPM is

estimated by the location obfuscation function $f(\acute{r}\backslash r) = Pr(\acute{r}\backslash r)$ relies on which one the mechanism replaces $r$ with $\acute{r}$, according to the probability distribution function. For a given user's real-location $r$, the obfuscation function $f(\acute{r}\backslash r)$ describes the probability that the protection mechanism produces an appropriate pseudo-location $\acute{r}$, ($\acute{r} \in \acute{R}$) as the output. The transformation decision can be seen as a defensive LPPM that specifies for each real-location $r$, a randomization over an achievable output of pseudo-location $\acute{r}$, which depends on the mobility profile $\psi(r)$. This is because continuously visited locations are identified, and each location can be related to private sensitive information. In the next section, we will describe how to precisely state the user's location privacy using the obfuscation mechanism $f(\acute{r}\backslash r)$.

### 3.3. *The Performance Metrics for Privacy Protection*

In the aforementioned LBS setting, there exist a number of LBS intuitive metrics for evaluating the performance of various protection mechanisms. As users want to measure their obtained service quality, resource utilization, and location privacy leakage when they access the LBS, they impose these qualitative evaluation metrics to evaluate and compare the privacy protection performances. These LBS performance metrics include an evaluation metric for measuring the service-quality loss and energy consumption incurred by using the LPPMs as well as an evaluation metric for quantifying the LBS privacy leakages (as an adversary's estimation error) under some location inference tracking attacks.

### 3.3.1. Quality of Service Analysis and Evaluation System

The user's LBS-related response relies on the LPPM-transformed pseudo-location $\acute{r}$, instead of the actual-location $r$. However, the distortion between the real-location $r$ and previously exposed location $\acute{r}$ determines the user's service quality loss. If the degree of similarity between the real and observed location traces is higher, the service-quality loss becomes lower. Therefore, the service-quality loss of the user provides a measure of how different the real location traces and the observation traces are. The Euclidean distance between the real-location $r$ and pseudo-location $\acute{r}$ can be denoted as the function $d_q : (\acute{r},\ r) \rightarrow d_q(\acute{r},\ r)$ to reflect the system's quality of service loss due to the distortion in the LPPMs. Thereafter, the expected-quality loss $Q_{loss}$ denoted by $E[Q_{loss}(\psi,\ f.\,d_q)]$ in the given LPPM function $f(\acute{r}\backslash r)$ can be estimated as:

$$E[Q_{loss}(\psi,\ f.\,d_q)] = \sum\nolimits_{\acute{r},r} \psi(r).f(\acute{r}\backslash r).d_q(\acute{r},\ r). \tag{1}$$

We assumed that the user experiences a maximum loss in the quality of service, denoted by $Q_{loss}^{max}$ if the LPPM uses the function $f(\acute{r}\backslash r)$ to substitute $r$ with $\acute{r}$. This is due to sharing the user's pseudo-locations or noisy data instead of the actual-locations. Then, the output of the pseudo-locations $\acute{r}$ generated by the LPPM function $f(\acute{r}\backslash r)$ must satisfy the following mathematical expression:

$$Q_{loss}(\psi,\ f.\,d_q) \le\ Q_{loss}^{max}. \tag{2}$$

From the above inequalities, the distortion brought in the previously observed (pseudo-locations) determines the service quality-loss constraint that each user experiences. We note that the impact of quality-loss $Q_{loss}^{max}$ on location privacy depends on the distance function $d_q(\acute{r}, r)$ and on the user's access to the LBS application type.

### 3.3.2. Energy Consumption Analysis

Mobile applications are commonly equipped with high resource-constrained smart phones and communication capabilities. The ability of these mobile applications to access LBSs relies on the users' cell phone battery as the energy supply, therefore, LBS users may want to consider the impact of privacy preservation on resource limitations. Compared with the existing LPPMs in Reference [18], the output of dummy and precision-based protection mechanisms may include more information

about transitions between regions to protect the user's privacy. This results in an increase in energy consumption when uploading and downloading the locations exposed by the LPPMs. In order to measure the cost of the energy constraints in different LPPMs, we extended the existing framework to describe that fact by defining the expected energy cost employed by the LPPMs. The distance between the actual-location $r$ and pseudo-location $\acute{r}$ can be denoted as the function $d_E(\acute{r}, r)$ to reflect the cost of energy by the user if $f(\acute{r}\backslash r)$ substitutes $r$ with $\acute{r}$ in the LPPMs to send and then receive the location data. Therefore, the expected energy cost $E_{cost}$, denoted by $E_{cost}(\psi, f. d_E)$ in the given LPPM function $f(\acute{r}\backslash r)$ that replaces $r$ with $\acute{r}$ is determined as:

$$E_{cost}(\psi, f.d_E) = \sum_{\acute{r},r} \psi(r).f(\acute{r}\backslash r).d_E(\acute{r},r). \tag{3}$$

We supposed that the user demands a maximum tolerable energy cost constraint resulting from the exposed pseudo-locations. Like the service quality loss constraint, the output of LPPM function $f(\acute{r}\backslash r)$ must satisfy the expression $E_{cost} < E_{cost}^{max}$.

### 3.3.3. Location-Privacy Leakage (Adversarial Estimation Error)

The adversary's objective is to speculate the users' location access profile and then infer their real location $r$ at a certain point and given time $t$. As the adversary's prior knowledge on users is the probability that the users are at certain points (selected location spots) when they use the LBS, the present locations of the users are conditionally independent of their previously observed and future locations. Given observations on the pseudo-location $r'$, user's profile $\psi(r)$, and transformation decision function $f(\acute{r}\backslash r) = Pr(r\backslash\acute{r})$, the adversary executes (runs) an inference attack on pseudo-location $r'$ to obtain estimations, $\hat{r}$, of the user's real location (or privacy information), $r$. Obviously, the adversarial estimation decision (equivalently, the user's privacy leakage) will be quantified by a loss function $d_p : (\hat{r}, r) \rightarrow d_p(\hat{r}, r)$ under the given LBS profile and pseudo-location observed by the adversary, which quantifies the adversary's loss caused by making decision $\hat{r}$ when the real location trace is $r$. The privacy leakage depends on both the location semantic diversity and the user's privacy criteria as well as the decision function $f(\acute{r}\backslash r)$ performed by the privacy protection mechanism [6,25]. Therefore, $d_p(\hat{r}, r)$ should be defined appropriately. However, if the mobile user wants to hide an exact present position around a large area of region, the appropriate distance function might be the discrete metric of $d_p(\hat{r}, r) = 1$ if $\hat{r} = r$, otherwise $d_p(\hat{r}, r) = 0$. Alternatively, the location privacy of the mobile user might rely on the physical distance (the squared-error distortion) between the real location $r$ and estimated location $\hat{r}$. In this case, the distance function can be equivalent to $d_p(\hat{r}, r) = (\hat{r}, r)^2$. In this spirit, we assumed that the adversary employed a Bayes decision estimator to identify the users' real location [30]. We used this decision to define the adversarial average loss as the Bayes conditional privacy risk for estimator $\hat{r}$, in the discrete case:

$$E\big[d_p(\hat{r},r)\big] = \sum_r d_p(\hat{r},r).Pr(r\backslash\acute{r}). \tag{4}$$

The adversary's goal is to choose $\hat{r}$ to reduce the location privacy of the users among all possible estimators, where the assumption taken under $Pr(r\backslash\acute{r})$ denotes the posterior probability distribution values for $r$, given the status of pseudo-location $\acute{r}$. Basically, the location privacy leakage depends on the privacy requirements of the user and on the attack strategy deployed by the attacker. In this case, the user strategically chooses an optimal obfuscation function $f(r'\backslash r)$ to maximize their location privacy based on the distance function $d_p(\hat{r}, r)$, and in situations where an adversary, based on their prior-knowledge, chooses the optimal location inference attacks $h(\hat{r}\backslash\acute{r})$ to reduce the maximum location privacy of the users under service quality constraints. Thus, the users' expected location privacy associated with the adversarial expected error is defined as the expectation of $d_p(\hat{r}, r)$, over all possible $r, \acute{r}, \hat{r}$ as follows [18]:

$$Privacy\left(\psi, f, h, d_p\right) = \sum_{\hat{r},\acute{r},r} \psi(r).f(\acute{r}\backslash r).h(\hat{r}\backslash\acute{r}).d_p(\hat{r},r). \tag{5}$$

Here, each summand combination of the equation establishes a probabilistic belief on each user's information that achieves a privacy amount of $d_p(\hat{r}, r)$, and the LPPM takes into account the successively visited locations (sensitive information of users) to increase the uncertainty for an adversary. However, the user's real location trace $r$ can be re-identified and observed by the adversary in an analysis given their LBS access and profiles ($\psi(r)$) and the obfuscation function $f(\acute{r}\backslash r)$ offered by a given protection system.

## 4. Privacy Game Formulation for Attack and Defense Mechanisms

In this section, we adopted a zero-sum Bayesian Stackelberg game model to precisely formulate the problems of an optimal protection mechanism against a strategic adversary who is aware of the users' access to the LBS pattern and the LPPM's internal algorithm to localize them [6,9]. In this game model, a user (i.e., as a leader) first chooses an optimal location obfuscation function $f(\acute{r}\backslash r)$ and runs it by considering the real location $r$ as the input and an observed pseudo-location $r'$ as the output. Then, the adversary (i.e., as a follower) plays next by running a location inference attack function $h(\hat{r}\backslash\acute{r})$ to estimate the user's location traces based on knowledge of the background information and obfuscation function $f(\acute{r}\backslash r)$. Both the user and adversary have targets in the privacy game model to maximize their payoffs under the cost of energy and quality of service-loss. This means that the user targets maximizing their location privacy while the adversary targets the reduction in this amount to improve the strategy and success of its attack. In this case, the privacy game depends on Bayesian probability because the location information of the user is not complete for the adversary. It is also a zero-sum game against inference attacks because each one's loss equals the other's gain.

We further extended the optimal protection and attack strategies of this game model [18] to obtain optimal dummy-and precision-based mechanisms that could be incorporated in the context of the existing framework. This enables all LBS users to find the optimal parameters to maximize their location privacy and quality of service against smart attacks that satisfy each user's requirements. More precisely, we next present the game problem statement for the user and adversary that we aimed to solve in this work. Thereafter, we estimated the classic formulation of the linear program to construct the optimal privacy protection and attack strategies, respectively.

### 4.1. The Problem Statement

This section precisely states the game problem statement and the assumptions for the LPPMs and adversary model, as illustrated in Figure 3. Given:

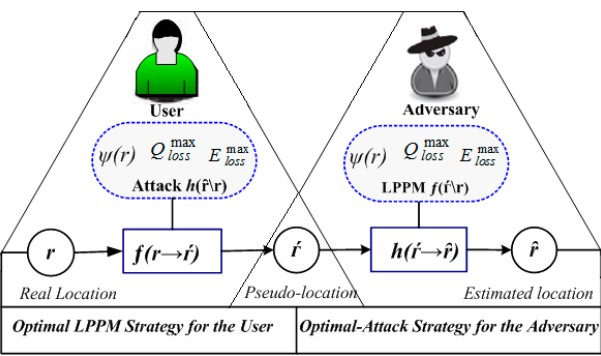

**Figure 3.** Information sharing between the user and adversary in the game model.

(a) The user's profile $\psi(r)$ or adversary's prior-knowledge acquired from the user's LBS access patterns, and from the successively observed locations of users,

(b) The maximum achievable quality of service loss $Q_{loss}^{max}$ experienced by the user as assigned for $Q_{loss}(\psi, f.d_q)$, this can be defined by the distance function $d_q(\acute{r}, r)$ between the real-location $r$ and pseudo-location $\acute{r}$, and

(c) The cost of the energy constraint $E_{cost}^{max}$ demand by the user's resource constrained mobile devices as assigned for $E_{cost}(\psi,\ f.\,d_E)$, this can be defined by the distance function $d_E(\acute{r}, r)$.

The purpose of this optimal solution is to determine the location obfuscation mechanism $f(\acute{r}\backslash r)$ that maximizes the expected privacy of the users as defined in Equation (5) and with our dummy-and precision-based mechanisms. Thus, the optimal solution should take into account that the adversary's observation on the user's profile $\psi(r)$ and output of the protection mechanism as well as the resource constraints in terms of quality-loss $Q_{loss}^{max}$ and energy cost $E_{cost}^{max}$ requirements. Figure 3 illustrates the information (resource) sharing between the user and the adversary in the game model, which takes into account the mobility profile of the users and resource constraints [9].

According to this expression, the adversary implements the location inference attack function $h(\hat{r}\backslash\acute{r})$ and pays an amount of the quality function $d_p(\hat{r},\ r)$, which indicates the location privacy (gained) for the users and equivalently the adversary's estimation-error. In order to examine the relationship between the user and the adversary under a common user-attacker game model, we need to define the procedures of the game equilibrium solution between them. This common knowledge to both the user and the adversary estimates the user's location privacy with the minimum quality-loss function of $d_p(\hat{r},\ r)$. Next we explain the linear programs used to construct the optimal protection and attack strategies with the inputs of $\psi(r)$, $d_q(\acute{r},\ r)$, $d_E(\acute{r},\ r)$, and $d_p(\hat{r},\ r)$, which compute the output of the optimal obfuscation function $f(\acute{r}\backslash r)$ for the users, and the adversary's inference attack $h(\hat{r}\backslash\acute{r})$, taking the service quality-loss into account. We note that, when we used the expression optimal obfuscation function $f(\acute{r}\backslash r)$ for the user and the adversary's optimal inference attack function $h(\hat{r}\backslash\acute{r})$, the two functions were optimal against each other; that is, we obtained a precise game-equilibrium solution for the problem [18].

*4.2. Optimal Privacy Protection for the User*

We assumed that an adversary could observe all of the distortion pseudo-locations output by the LPPMs and could form the posterior distribution value on the user's actual location $r$, conditional on observation $r'$ [9,18]. The posterior probability distribution of the real location $r$ can be computed by the adversary by exploring the background information of the user and the previously observed pseudo-location trace $\acute{r}$. based on Bayesian probability, the adversary can use this rule to induce a posterior distribution on the user's private information as follows:

$$Pr(r\backslash\acute{r}) = \frac{Pr(r,\acute{r})}{Pr(\acute{r})} = \frac{f(\acute{r}\backslash r)\psi(r)}{\sum_r f(\acute{r}\backslash r)\ \psi(r)} \tag{6}$$

where $Pr(r\backslash\acute{r})$ denotes the posterior probability values for the real location $r$, given the status of the observed pseudo-location $\acute{r}$. The objective of the adversary is then to choose $\hat{r}$ to reduce the conditional expected privacy of the user where the posterior probability distribution $Pr(r\backslash\acute{r})$ is taken under expectation. Given an arbitrary estimator $\hat{r}$, the user's conditional expected privacy in the discrete case is computed by:

$$\sum_r d_p(\hat{r},r).Pr(r\backslash\acute{r}). \tag{7}$$

Thus, the probability that the privacy preserving-mechanism produces the output $r'$ is defined as

$$Pr(\acute{r}) = \sum_{\acute{r}}\psi(r).f(r\backslash\acute{r}). \tag{8}$$

Therefore, the user's unconditional location privacy associated with the adversarial average loss as defined in Equation (4) in their inference attacks over all possible observation $r'$ can be expressed as:

$$E\left[d_p(\hat{r},r)\right] = \sum_{\acute{r}} Pr(\acute{r})min_{\hat{r}}\sum_r d_p(\hat{r},r).Pr(r\backslash\acute{r})$$
$$= \sum_{\acute{r}} min_{\hat{r}}\sum_r \psi(r).f(r\backslash\acute{r}).d_p(\hat{r},r) \tag{9}$$

where the expectation is taken under the joint probability distribution of $r$ and $r'$, the adversary's optimal strategy is to select $\hat{r}$ to reduce the privacy of the users among all possible estimations. To facilitate the computations, we applied Equations (6) and (8) to Equation (9) and the minimized unconditional location privacy of the user can be expressed as:

$$(Pr_{min}(\acute{r}) = x_{\acute{r}}) \triangleq \sum_r \psi(r).f(r\backslash\acute{r}).d_p(\hat{r}, r). \tag{10}$$

The user takes the adversary's background knowledge into account for the chosen protection mechanism $f(\acute{r}\backslash r)$, which is anticipated to maximize the location privacy of the users while ensuring the requirements of service quality-loss. The linear program to determine the optimal parameters for the user with an objective of maximizing their privacy, which corresponds to Equation (10), is as follows. Then, the user chooses the function $(f(\acute{r}\backslash r), x_{\acute{r}}, \forall r, \acute{r})$ to find $f(\acute{r}\backslash r)$ where

$$\text{Maximize}: \sum_{\acute{r}} x_{\acute{r}} \tag{11}$$

$$\text{Subject to}: x_{\acute{r}} \leq \sum_r \psi(r).f(\acute{r}\backslash r).d_p(\hat{r}, r), \forall \hat{r}, \acute{r} \tag{12}$$

Here, the variable $x_{\acute{r}}$ is a determination variable in the optimization problem, and integrating the user's service quality-loss and energy cost constraints, we get:

$$\sum_r \psi(r) \sum_{\acute{r}} f(\acute{r}\backslash r).d_q(\acute{r}, r) \leq Q_{loss}^{max} \tag{13}$$

$$\sum_r \psi(r) \sum_{\acute{r}} f(\acute{r}\backslash r).d_E(\acute{r}, r) \leq E_{cost}^{max} \tag{14}$$

Under the users' required service quality and energy cost constraints, the probability distribution function is also shown as follows:

$$\sum_{\acute{r}} f(\acute{r}\backslash r) = 1, \text{ and } f(\acute{r}\backslash r) \geq 0, \forall r, \acute{r}. \tag{15}$$

The inequalities defined by Equation (12) represent the privacy restraints, which show that $f(\acute{r}\backslash r)$ is preferred to maximize $x_{\acute{r}}$. Next, the inequality denoted by Equation (13) represents the quality-loss constraint, which guarantees that the expected service quality loss is at most $Q_{loss}^{max}$. The newly added inequality equation defined in Equation (14) represents the energy cost constraint. The main goal of this setting ensures that both the expected energy and service-quality consumption correspond to the dummy-and precision-based mechanisms that do not exceed their thresholds $E_{cost}^{max}$. The last expression represented by Equation (15) guarantees that $f(\acute{r}\backslash r)$ is a proper probability-distribution.

When using both the dummy and precision-based LPPMs, the outputs of the exposed pseudo-locations $\acute{r}_i$ correspond to the optimal LPPM function $f(\acute{r}\backslash r)$ from the mobility profile $\psi(r)$ belonging to $\acute{R}_i = P(\acute{R}) - \{0\}$. The user requests the dummy queries first and then the LPPM generates a fake or noisy-version of their locations to improve the adversary's estimation error, but it may or may not include the user's actual location (i.e., $r_i \in R$). In dummy-based LPPMs, the pseudo-locations contained in $\acute{r}_i$ are produced by a group of one or multiple non-contiguous locations, however, in the case of precision-based LPPMs, it is generated by a set of one or multiple successive contiguous locations (i.e., $r_i \in R$). Compared with the perturbation-based LPPM in [18], the outputs of the LPPMs corresponding to the dummy and precision-based strategies may contain a set of multiple regions to conceal the location information or service-query attributes. In this case, the system will send and receive more location data to/from the LBS provider when performing the LPPM, which will result in the consumption of more energy. Therefore, the dummy-based or precision-based mechanisms only perturb the previously exposed locations as each pseudo-location output by the LPPMs may or may not include the user's real locations.

### 4.3. Optimal Inference Attack for the Adversary

Similar to the optimal protection strategy for the users, we analyzed the adversary's strategy where an inference attack is executed on the pseudo-location $r'$ to obtain the estimations, $\hat{r}$. The adversary makes use of all of the distortion pseudo-location $\acute{r}$ outputted by the LPPMs to determine the estimator $\hat{r}$ that solves Equation (15). Given the real location $r$, pseudo-location $r'$, and conditional probability $Pr(r\backslash\acute{r})$, the conditional expected location privacy of the users is expressed as

$$\sum_{\acute{r}} Pr(\acute{r}) \sum_{r} d_p(\hat{r}, r).Pr(r\backslash\acute{r}) \tag{16}$$

where the joint probability distribution of $r$ and $r'$ are taken as expectations. However, the user chooses $\acute{r}$ to maximize Equation (16), while the adversary implements the location inference attack function $h(\hat{r}\backslash\acute{r})$ to minimize the privacy of the users when they use the location obfuscation $f(\acute{r}\backslash r)$. In this case, we used the variable shadow price $z_q$ to model the relationships between privacy and service quality [18]. Therefore, given the attack function $h(\hat{r}\backslash\acute{r})$, the expected-privacy ($y_r$), the privacy minimization weight coefficients $z_q$ and $z_E$ for $Q_{loss}^{max}$ and $E_{cost}^{max}$ (for all $r$, $\acute{r}$ and $\hat{r}$, and $z_q$& $z_E$ ∈ [0, ∞)), we can obtain the function $h(\hat{r}\backslash\acute{r})$ to reduce the user's maximum location-privacy under the quality loss constraint and we can assume that $y_r$ is the expected-privacy for the user. Then we obtain:

$$\text{Minimize}: \sum_{r}\psi(r).y_r + z_q.Q_{loss}^{max} + z_E.E_{cost}^{max} \tag{17}$$

$$\text{Subject to}: y_r \geq \sum_{\hat{r}} h\,(\hat{r}\backslash\acute{r}).d_p(\hat{r}, r) - \left(z_q.d_q(\acute{r}, r) + z_E.d_E(\acute{r}, r)\right), \forall r,\,\acute{r} \tag{18}$$

$$\text{Finally, we obtain}: \sum_{\hat{r}} h\,(\hat{r}\backslash\acute{r}) = 1, \forall \acute{r},\text{ and } h(\hat{r}\hat{r}\backslash\acute{r}) \geq 0, \forall \acute{r}, \hat{r}\,\hat{r} \tag{19}$$

$$\text{We assume that}: z_q \geq 0 \text{ and } z_E \geq 0 \tag{20}$$

In the above equations, $y_r$ describes the expected-privacy of the user and variable $z_q$ serves as the *shadow-price* for the service quality constraint, which can be considered as the "exchange rate" between privacy and service quality [18]. It describes the privacy gain (loss), when the maximum acceptable loss quality of service $Q_{loss}^{max}$ increases (decreases) by a unit price of quality. The minimization-term defined by inequality (Equation (17)) consists of weight coefficient variables $z_q$ and $z_E$, correlating to $Q_{loss}^{max}$ and $E_{cost}^{max}$ accordingly. The value of these weight coefficients indicates the degree of privacy gain (loss) with the increase or decrease of threshold values $Q_{loss}^{max}$ and $E_{cost}^{max}$. The distance functions $d_q(\acute{r}, r)$ and $d_E(\acute{r}, r)$ model the relationship between privacy, quality-loss, and energy cost requirements. The last expression (Equation (20)) provides the balance between privacy and the constraints computed by the adversary with weight coefficients $z_q$ and $z_E$ are always positive.

## 5. Performance Evaluation and Results

This section presents our theoretical analysis to evaluate the effectiveness of the optimal strategies and compare the performance of the existing as well as proposed LPPMs. We justified the proposed optimal privacy protection parameters including the experimental settings of real mobility traces, the impact of service-quality and energy constraints on location-privacy, as well as the balance between location privacy and the constraints on the quality of service and energy consumption. The proposed optimization model evaluates the impact of the service quality loss and energy consumption analysis presented in Sections 4.2 and 4.3 on the privacy protection guarantees with different LPPMs. Before reporting the performance results, we first introduce the experimental-setup such as device configuration and parameter settings.

## *5.1. Parameter Settings and Configuration*

In order to evaluate the performance of the proposed protection mechanisms, we used real mobility profiles obtained from publicly available crowd sensing datasets. Crowd sensing dataset projects that report a large number of wireless network research data, which are deposited in a publicly available database for the research community, specify the locations of the deposited data traces and provide the relevant materials for collecting and analyzing the data. These datasets contain the users' real location traces at a high granularity. There exist several crowd sensing datasets that are publicly available in real people mobility traces. For example, there are various real mobility traces of student life on campus [31], entertainment and sporting activities [32,33], and of taxis cabs [34,35].

In this paper, we used GPS-based real mobility traces of taxi cabs obtained from the CRAWDAD dataset project [35], which contains the GPS location coordinates of around 320 taxis collected within one month (30 days) in the city of Rome, Italy. This project dataset contains community life routing, transportation, as well as sports and entertainment activities. Each taxi driver uses a GPS-enabled mobile phone that updates the current position and sends the updated location to a central server in the interval of approximately seven seconds. Since the adversary implements the attack function $h(\hat{r}\backslash\hat{r})$, it is assumed that they obtain access of the user's mobility profile as well as the previously observed locations, from which the user accesses their LBS at a given time. Therefore, the first procedure is to mine their points-of-interests (POIs) according to the datasets in order to provide the adversary with the background information for speculating and then inferring the users' historical traces [36,37]. For this purpose, we used the parameters in terms of interest area size and the number of regions to construct the user's LBS access pattern and mobility profile model. For the sake of simplicity, we limited our analysis to the dataset of mobility traces, where the density of the taxi cabs is relevant. We randomly chose five mobile users and focused on their location traces during the morning and afternoon when the user was more likely to use LBSs. The quantitative experiments were run and implemented in the MATLAB platform software. The machine was equipped with a Lenovo, Intel(R) Core(TM) i5-2400 CPU, Quad @3.10 GHz, Windows 7 Ultimate, 64-bit operating system equipped with a 4 GB installed memory (RAM). The quantitative parameter settings and estimated values are shown in Table 2. In the next section, we describe the number of available regions and the selection for the area size of the POIs used in our experiments. The parameters used in the evaluation are summarized in Table 2 based on the real mobility dataset included in [35].

**Table 2.** Summary of quantitative parameters.

| Parameters | Value |
|---|---|
| Real mobility traces of taxi cabs | 320 |
| User's moving area | 8 km × 10 km ($\approx$ 80 km$^2$) |
| Total Number of regions | 20 × 16 (= 320) |
| Most considered regions | 20 |
| Number of users selected | 5 |
| Length of considered times | 1 h, 2 h, 3 h |

## *5.2. Modeling the Location Mobility Trace Based on POI Mining*

The proposed protection mechanism used in this work is a combination of the POI mechanism and the location obfuscation mechanism provided in Reference [22] and Reference [38], respectively. In order to evaluate the statistical relevance of the real location traces across all the regions, we began with the POI-based mechanisms and attacks to obtain access to the user's mobility profiles in the datasets. The method of the user's location trace distribution based on POI mining was introduced by the author in Reference [38] from the GPS location dataset. This mechanism was implemented to represent the set of stay points where users may perform different activities. Recalling that each point of interest grid describes a minimum particle size location that is located at the center of each cell, therefore, a smaller number of points of interest describe larger cells. We considered the user's location

area of $10 \times 8$ [km $\times$ km] = 80 km$^2$; the area within which they move as divided into 320 available location traces (regions). Figure 4 indicates the users' location distribution density in different part of regions. In particular, each region describes the distribution of the users' location in each particle size area, which corresponds to the granularity of the minimum size positions.

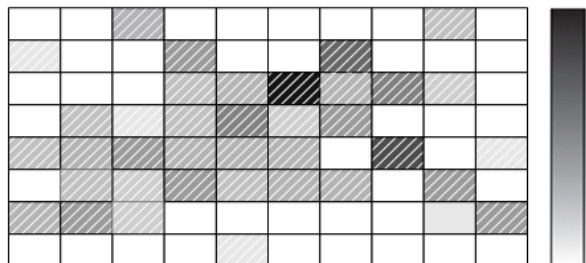

**Figure 4.** Spatial-histogram illustrates the density of the users' distribution per minimum particle size of regions in log-scale. Considered moving area = 80 km$^2$, divided into $10 \times 8$ regions.

The grid size can be estimated by considering the interest area divided with a $M = \alpha \times \beta$ grid of cells (regions), and here, there is no particular limitations on the size or shape of the regions. From the above representation, the grayness of each cell in the region indicates the density of its equivalent position in log-scale. A darker region describes a higher probability of the user to access the LBSs from that location. On the other hand, the mobility profile of each user describes the normalized set of their regular visits to a certain location. We recall that the user maximizes their location privacy and wants to run the linear program solution once to estimate their protection mechanism. The user's mobile device can contract out this operation to a service provider through an adequately secured connection. A larger number of regions can be considered in order to secure a higher-level of user privacy.

*5.3. Optimal Protection and Attack Strategies*

In the obfuscation-based mechanism, the real location trace of each past revealed location can be replaced by a pseudo-location in the set of regions $\acute{R} = \{\acute{r}_1, \acute{r}_2, \dots, \acute{r}_M\}$. Each possible pseudo-location $\acute{r}$ output by the obfuscation-based mechanism correlates to a subset of positions in $R$ (i.e., $\{\acute{r} \in \acute{R}\}$). For this purpose, we used a prior work [18] as a reference and constructed the basic obfuscation function level $k = 1, 2, 3, \dots$, for each real location $r$, and then we defined its distribution by the $k - 1$ nearest locations. In this case, the distribution decision function shown in Equation (6) will be a uniform distribution on the set of the $k - 1$ possible pseudo-location including the real location. The real location $r$ is substituted by a $k$-pseudo-location, with uniform probability $1/k$ and all the remaining locations have a probability of 0. As the focus was on optimal sporadic LBS in which the locations from successively visited LBS subsequent requests were not correlated, we limited our analysis to the present locations and not on past exposure. In order to evaluate the expected service-quality loss $Q_{loss}$ for any obfuscation-based LPPM, we used Equation (2) whether it was optimal or not.

The optimal dummy-based mechanism allows for the outputs of exposed locations to include any combination of noncontiguous locations, from which we can then estimate the number of potentialities corresponding to a subset of regions. The existing dummy-based protection mechanism can be modeled as follows: The LPPM allows the user to set a value for the constraints on additional resource cost that establishes the allowed resource consumption. Then, the outputs of the exposed pseudo-location $\acute{r}_i$ is chosen based on the LBS user's profile from all possible obfuscated locations that include a set of multiple regions. For precision-based mechanisms, only the previously exposed locations are perturbed; the real location may or may not be perturbed. Furthermore, we analyzed the performance of optimal and existing protection mechanisms against inference attacks as described in Sections 3.2 and 3.3.

*5.4. Evaluation Results*

The quantitative evaluation model enables us to determine the effectiveness of optimal protection mechanisms against inference attacks in the LBS to ensure the satisfaction of quality of service and resource consumption. We present the evaluation results through the following components. First, we show that the two popular solutions such as dummy-and precision mechanisms modeled using the existing framework are superior to the state-of-the-art protection mechanisms. Then, we analyze the impact of resource constraints (i.e., service quality loss and energy consumption issues) on the privacy protection granted by the optimal mechanisms. Finally, we compare the effectiveness of the privacy protection effects, aiming at the relative importance of precision and dummy-based mechanisms that generate dummy queries or reduce the precision of the pseudo-locations. For this purpose, given the specific quantitative parameters, which are described in Table 2, the optimal protection and its corresponding novel tracking attack were computed using the linear programming by solving Equations (11) and (17).

5.4.1. Privacy Protection under Quality-Loss Constraint

For the sake of performance reasons, we first started the evaluation analysis on the existing perturbation-based mechanism used in Reference [18] by using real location traces. In the first experimental analysis, we observed the impact of tolerated quality of service loss on location privacy protection and, more importantly, to see the relationship between the service-quality threshold and quality-loss when obtaining optimal privacy. As shown in Figure 5a,b, the protection mechanism was designed against the optimal attack obtained by the optimization problem for various service quality constraints. We observed that the optimal location privacy value increased with service-quality constraints to a maximum value and then remained unchanged while a particular privacy value was reached. Figure 5c,d describes the relationship between service-quality and maximum tolerated loss of quality of service $Q_{loss}^{max}$ when attaining the expected location privacy for the user. We observed that once the desired privacy level was reached, releasing the service quality requirements may not significantly affect the privacy protection results. On the other hand, the maximum amount of privacy obtained by the optimal protection mechanism relied on the level of the quality-loss constraint and on the user's mobility profile. This confirms that the previous results in Reference [18] demonstrated the tradeoffs between service-quality and privacy constraints for a given combination of service-quality threshold that have the same tendency for different users.

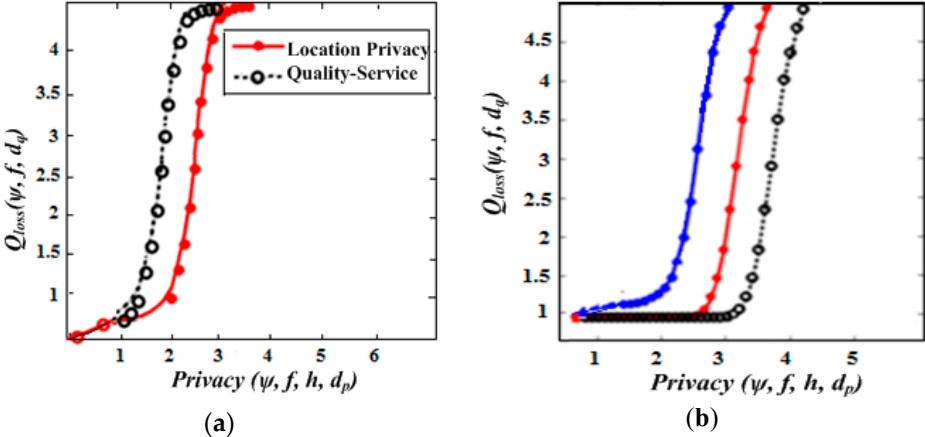

**Figure 5.** *Cont.*

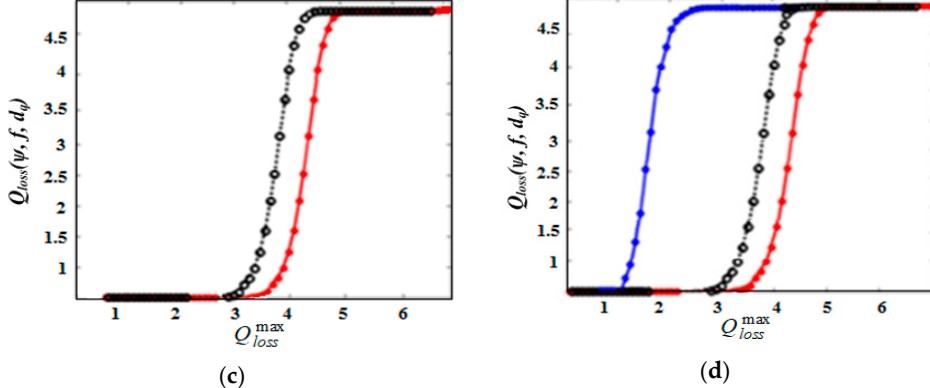

**Figure 5.** Impact of service-quality requirements $Q_{loss}^{max}$ on optimal location privacy. (**a**) Privacy value against inference attack, and expected service quality. (**b**) Privacy vs. service quality loss for a given $Q_{loss}^{max}$. (**c**) Privacy and Service quality-tradeoffs. (**d**) Service-quality threshold–Service-quality loss.

From Figure 5, we also observed the service quality requirements and the corresponding privacy obtained as the optimal protection mechanism, which was configured with increasing maximum values of the tolerated expected quality of service $Q_{loss}^{max}$. Therefore, the adversary's estimation was only dependent on their previous experience and background information on the user's location-distribution pattern and mobility profile. This is because losing the quality of service requirements allows the LPPM to select the exposed observed-locations that do not leak useful location data, which is beneficial for the adversary to limit the privacy protection. As a result, the expected service quality requirement increases slowly and loses the optimal expected privacy value, which can be lower than the maximum achievable service quality threshold value $Q_{loss}^{max}$.

### 5.4.2. Performance of Existing and Optimal Strategies

The purpose of the experimental studies in this section was to analyze the characteristics of the service quality constraints that allow for the protection mechanisms to perturb the real location and energy consumption when obtaining the optimal expected values, and more importantly, to analyze the location-privacy advantage of the optimal parameters against inference attacks to reduce quality-loss. Given the users' location in each grid size, we observed in the level of privacy, when the resource consumption was enabled, that both the existing and optimal mechanisms reached the maximum amount of privacy feasible. The theoretical analysis justified that when the expected quality of service requirement was losing, both mechanisms were permitted to add a large amount of noise so that the adversary's observation was very uncertain and there was nothing special about the real location of the user. On the contrary, when the average expected quality loss was tight, the optimal parameter produces the limited noise allocation of the LPPM that it is allowed to inject [6]. Figure 6 shows the results of different LPPMs in the given values of quality-loss and energy cost threshold as a function of the LPPMs, when obtaining the expected location privacy. Figure 6a recalls that when the optimal expected privacy was reached, further releasing the service quality requirement slows the growth of the maximum achievable quality of service loss $Q_{loss}^{max}$, and does not result in a gain in privacy. Figure 6b–d shows the results under the precision-and dummy-based mechanisms for the user when $Q_{loss}^{max} = 0$. The relationship between the service-quality threshold and quality-loss under the dummy-and precision-based LPPMs under varying $E_{cost}^{max}$ are shown in Figure 6e,f. As a consequence, our evaluation analysis focused on the case where $Q_{loss}^{max} = 0$ so that the characteristic of quality loss requirement did not allow for the perturbation mechanism and the LBSs remained useful for the user.

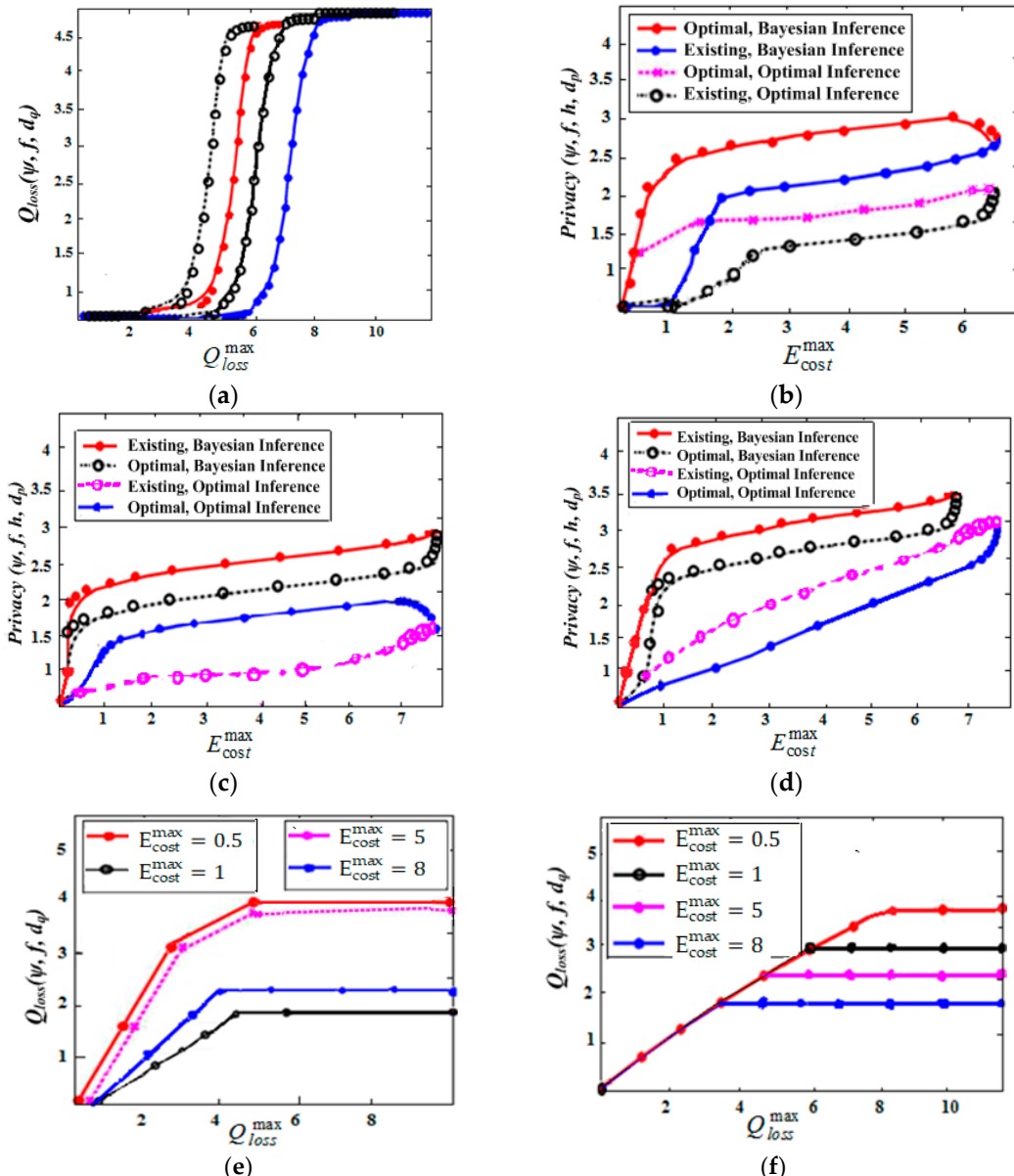

**Figure 6.** Effectiveness of the optimal dummy-and precision-based location privacy-preserving mechanisms (LPPMs) against inference attacks. (**a**) Expected quality loss under a given privacy level. (**b**) Existing and optimal LPPM against optimal attacks. (**c**) Dummy-based mechanism $(Q_{loss}^{max} = 0)$, no perturbation. (**d**) Precision-based mechanism $\left(Q_{loss}^{max} = 0\right)$. (**e**) Quality loss in Dummy-based LPPM. (**f**) Expected quality loss in precision-based LPPM.

From Figure 6, we observed that when the optimal expected privacy level was reached, the service quality requirements of the precision protection mechanism was smaller than that of the dummy mechanism, more importantly because a higher degree of freedom was chosen in the dummy-based mechanism to hide the locations when compared to the precision-based mechanism. In order to improve the privacy protection effects, the dummy-based protection strategy chooses an observed location far away from the true location, which gives larger quality of service-loss constraints. Compared with the perturbation-based framework in Reference [18], the service quality constraints of the precision-based mechanism at the optimal privacy value increased with $Q_{loss}^{max}$, which increased linearly first, and then tended to remain unchanged. Therefore, once a particular amount of privacy was obtained, then there was no resource consumption in the system framework. Consequently, the cost of energy constraint did not affect the optimal attack and defense solutions and cannot be computed in linear programs.

### 5.4.3. Effectiveness and Advantage of Optimal Protection

Consider the mobile users making use of the optimal protection mechanisms against inference attacks on their mobile devices. When using the quantitative evaluation metrics for evaluating the effectiveness of the optimal LPPMs, we achieved the optimal expected privacy given the user's privacy leakage (equivalently, the adversarial estimation error) modeled by the probability distribution over the user's real locations (privacy information) and more importantly, to ensure a good balance between location privacy and the constraints on service quality and energy consumption. In the previous subsections, we analyzed the impacts of service quality and energy constraints at the optimal expected privacy, where the focus was on solving the optimization problem as a location obfuscation function and inference attack strategies to reduce quality-loss. Figure 6 shows the results of the optimal dummy-and precision-based LPPMs against inference attacks, when achieving the optimal expected location privacy. Then, the service quality threshold $Q_{loss}^{max}$ value was set to find the optimal parameters to reduce the service quality-loss.

Figure 6c,d shows the impact of the service-quality threshold $Q_{loss}^{max}$ on privacy for the precision and dummy-based protection mechanisms. Figure 6e,f illustrates the impact of the energy constraint $E_{cost}^{max}$ on location privacy, and primarily, to analyze the relationship between the service-quality threshold and quality-loss under the dummy-and precision-based optimal strategies. For a given LBS user and tolerable loss quality of service, the level of achievable privacy increased with an increase in the $Q_{loss}^{max}$ when the user imposed the optimal strategy. We noted that there was a possibility to reduce the quality-loss by increasing the average expected service quality requirements due to perturbations in the previously exposed locations, which is more easily performed by the precision-based mechanism. The reduction of this constraint may improve the expected privacy in particular quality-loss requirements, rather than privacy redaction. From the above analysis, it can be concluded that only loosening the energy constraint will not be sufficient to improve privacy, which largely depends on the amount of the service quality constraint. As expected, if energy consumption is allowed ($E_{cost}^{max} = 0$), the expected privacy increases by the amount of service quality constraint due to perturbations in the LBSs implemented by the LPPM. Furthermore, under a given combination of service quality and energy constraints, the dummy-based mechanism provided better achievable privacy protection with a limited resource-utilization. The conclusion in all adequate combinations of both constraints is that the dummy-and precision-based optimal strategies are preferred for a fine-grained comparative evaluation of the effectiveness of the various LPPMs.

### 6. Conclusions

The wide use of LBS applications when using smart mobile devices has come with new privacy risks for users. In this paper, we presented an extended user-centric privacy-preserving framework that allows the LBS user to specify the impact of quality-loss and energy constraints. Based on this, we formalized the optimal precision and dummy-based LPPMs by extending the optimal solutions of existing perturbation-based LPPMs while focusing on the impact of reducing the precision of the location information shared with the service provider and reporting location dummies to the LBS. Furthermore, we realized the strategy of privacy protection against location inference attacks by extending the classic formulation of the game theory model.

We validated the performance of the optimal location obfuscation function with respect to inference attacks by using the real mobility datasets, indicating that the optimal expected privacy was closely related to the impact of service quality and energy cost constraints. The results showed that the optimal precision-and dummy-based mechanisms performed better than the existing perturbation-based strategies in all adequate combinations of service quality and energy constraints. The evaluation results contribute to a better understanding of the advantages of precision- and dummy-based mechanisms in the context of the existing framework and to the design and evaluation of efficient LPPMs under service quality and resource utilization requirements.

**Author Contributions:** M.K.T. proposed the idea of the research, prepared the original draft data, analyzed, and performed the experiment analysis. X.Y. finalized the structure, reviewed, and edited the final paper.

**Funding:** This research was funded in part by the National Natural Science Foundation of China (NSFC) under Grant No. 61671057.

**Acknowledgments:** The authors wish to thank the School of Computer and Communication Engineering at University of Science and Technology Beijing. We also pleasantly acknowledge the financial assistance and support provided by NSFC.

**Conflicts of Interest:** We declare that there is no conflict of interest in our research article.

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
