# Peer review of "Location Privacy Protection Systems in Presence of Service Quality and Energy Constraints"

_information, doi:10.3390/info10040121_

Round 1

Reviewer 1 Report

The article faces the problem of analyzing user-centric location privacy preserving mechanisms (LPPMs) against location inference-attacks under service quality and energy constraints.This problem is challenging since users may demand heterogeneous service quality and energy constraints according to the context in which they are involved in. Also, there are many LPPMs, which may cause different scalability issues at the user and LBS side.

The problem discussed by this article is relevant since since users are becoming more aware of their privacy when accessing the Internet from a mobile wireless network.  This paper extends an location privacy existing framework [22] and it models and evaluates different LPPMs as the precision and dummy based techniques. However, the paper presents several issues which are listed as follows:

Article has a few English typos and some sentences are hard to read, please go over again.

Existing dummy based approaches assume a k-anonymity value, which is always greater than 1. This means the exact location of the user observed at time t (r) is replaced by  k dummy locations and one of these locations is r. It is unclear how  a distance error function is  computed between  k-1 dummy locations against the real one, how was the value for K defined?. I think, authors may add a few simple example how the computation is done based on Figure 2 and  give some details about the implemented mechanisms.

Ben Niu's articles show that k-dummy locations are chosen in such way that  all k locations are equally likely (theoretically) to be the real location (r). In order to achieve this goal, their techniques intend to maximize the entropy of the selected set of dummy locations. How do the tested dummy mechanisms consider this goal and how is this (or not) related to the function Ψ?

The curves displayed at Figures 5(b) and 5(c) (the evaluation result section) should be properly labeled. It is difficult to infer, which mechanism corresponds to each curve

Most of the paper is based on reference [22] and because of  this, it is unclear how the this article distinguishes from this cited reference in order to properly determine the novelty of the article. I suggest, authors may describe a final and a short paragraph at the end of the related work section, explaning the main differences between this article and the cited ones, specially [22].

Author Response

Dear Reviewer:

Thank you very much for reviewing our manuscript. We have implemented your comments and suggestions to revise our original submission. Please find our point-by-point response (in Red) to the specific comments (in black) in the file attached.

   Thank you very much!

          Sincerely,

                 Dr. Mulugeta Kassaw On behalf of co-authors

Reviewer 2 Report

While the paper addresses an interesting and relevant problem (namely, privacy protection in LBSs), it is quite difficult to understand as the writing is not clear. The paper needs substantial revision for clarity before it can be adequately reviewed.

Some specific comments:

I am unclear on several sentences within the introduction section (for example., '...by running a multiple of locations from where various consecutive queries have reported by mobile users."). The entire paper needs to be reviewed and revised for clarity; otherwise, it is difficult to ascertain what the issues are that are being addressed.

In the related work section, there needs to be differentiation between what the user does and what is programmed in to the specific LBS service.

The related work section also needs to identify some of the issues faced by existing methods of obfuscation, etc. Triangulation of locations, issues in sparse networks, etc. should be acknowledged. Without this, the need for the proposed framework is less clear.

The framework needs to be more clearly described. The discussion on p. 4, lines 170-178 isn't completely clear - what is the user's mobility model? Who are the actors (in the description, you seem to have mixed the user, the service, and the attacker - these need more differentiation)? It would be useful to have a more generalised overview of the entire framework to better contextualise the following discussion.

How well does the framework work if N is small?

Are there any provisions for use of multiple LBS apps?

It would be useful to have a more comprehensive summary provided that identifies benefits of the proposed system and potential limitations (particularly when compared to the other approaches reviewed).

Author Response

Dear Reviewer:

Thank you very much for reviewing our manuscript. We have implemented your comments and suggestions to revise our original submission. Please find our point-by-point response (in Red) to the specific comments (in black) in the file attached.

Thank you very much!

Sincerely,

  Dr. Mulugeta Kassaw on behalf of co-authors

Round 2

Reviewer 1 Report

In my understanding, authors obfuscates the real location of each past revealed location by a  k-pseudo location following a prior work [18] of other authors. In this work, Shokri et al, follow this idea: "For each location r, we find its k−1 closest locations (using the Euclidean distance between the centers of the regions... an actual location r is hidden among its k−1 nearest locations". Then Shokri et al study when they vary "the obfuscation level from 1 (minimum) to 30 (maximum), and for each case we compute the corresponding quality loss". I think you follow the same idea but you should indicate the level of obfuscation  (which is the parameter k) used  in your experiments. Please indicate this details in your draft.

I think, equation (19) has some errors. Please check it. There are still some typos like in lines 185, 471, 477, please proofread the paper again. Please  check font size of the legends for figures 6(e) and 6(f).

Author Response

 Dear Reviewer

Thank you for your helpful comments and suggestions. We have revised our paper accordingly and feel that your comments helped clarify and improve our paper. Please find our response in the file attached.

  Thank you with best regards,

       Dr. Mulugeta Kassaw

Reviewer 2 Report

The following sentence (P. 2 of 21, lines 88-89) is not clear and needs revision: “First, we justified an existing user-centric privacy framework to evaluate the privacy of users in the LBS to ensure the satisfaction of the quality of service and energy consumption.” The following sentence (P. 6, Lines 216-218) is a fragment and needs revision: “We considered that mobile users who want to protect their location information and service attributes from malicious attackers who can observe the users’ locations and background information.” The following sentence (P. 8, lines 290-292) is awkward and in need of revision: “The ability of these mobile applications to access LBSs relies on the users’ cell phone battery as the energy supply, therefore, users need to apply the LPPMs and then modify their location data to preserve their location privacy.” The order of operations to preserve location privacy isn’t clearly aligned with battery requirements. You state (P. 8, lines 308-310) that: “As the adversary’s prior knowledge on users is the probability that the users are at certain locations when they use the LBS, the present locations of the users are conditionally independent of their previously observed and future locations.” Is this accurate if traces (or paths) rather than points are being collected? Depending on the spatio-temporal nature of the data, it would seem that knowing the previously observed locations could reveal the current location. The following statement (P.9, lines 369-374) needs revision for clarification: “Given (i) the user's profile ψ(r) or adversary’s prior-knowledge acquired from the previously observations, (ii) the maximum achievable quality of service loss ????? ???experienced by the user as assigned for loss?ψ, f. dq?, defined by the distance function dq(ŕ,r) between the real-location r and pseudo-location ŕ, and (iii) the cost of the energy constraint ????? ??? demand by the user’s resource constrained mobile devices as assigned for Ecost(ψ, f. dE ), defined by the distance function dE(ŕ, r).” P. 9, lines 376-379 also needs revision.

Author Response

Dear Reviewer We would like to thank you for your detailed comments and suggestions for the manuscript. We believe that the comments have identified important areas which required improvement. Please find our point-by-point responses in the file attached, Thank you very much for your effort.    Yours sincerely,  Dr. Mulugeta Kassaw on behalf of my co-authors

Round 3

Reviewer 1 Report

I think authors have addressed the comments indicated to them. I think this paper is now ready to be published in the journal

Author Response

(The authors gave the same response as above.)
